# Vascular Abnormalities of the Parotid Region: An Uncommon Presentation of a Common Condition—A Case Series

**DOI:** 10.3390/diagnostics12092236

**Published:** 2022-09-16

**Authors:** Tomasz K. Nowicki, Michał Joskowski, Ewa Garsta, Bogusław Mikaszewski

**Affiliations:** 12nd Department of Radiology, Faculty of Health Sciences, Medical University of Gdansk, Smoluchowskiego 17, 80-214 Gdansk, Poland; 2Department of Otolaryngology, Faculty of Medicine, Medical University of Gdansk, Smoluchowskiego 17, 80-214 Gdansk, Poland

**Keywords:** vascular malformation, thrombosis, parotid gland, parotid region

## Abstract

A variety of non-neoplastic diseases and benign and malignant lesions may involve parotid glands. In clinical practice, effective diagnosis is crucial to ensure proper treatment and achieve a good therapeutic effect. Unclear anamnesis and short medical history are factors that make diagnosis difficult, especially when cancer should be excluded. We present a case series of four patients who reported to the outpatient clinic with a unilateral nodule in the parotid region. The clinical presentation prevented an unequivocal diagnosis. The suspicion of a neoplastic disease resulted in profound diagnostics, including repeated cytology, ultrasound and magnetic resonance examination. Combining all the acquired information and follow-up, or a histopathologic examination, facilitated the final diagnosis. In all cases, thrombosis was diagnosed. We then analysed the diagnostic process and the associated difficulties. When thrombosis in vascular malformation occurs in the parotid region, it may have an unclear clinical and radiological presentation. Such an image can imitate both benign and malignant tumours. Ambiguous imaging in conjunction with blood cells in cytology should result in the inclusion of thrombosis in vascular malformations in the differential diagnosis.

## 1. Introduction

Vascular malformations (VMs) belong to vascular abnormalities resulting from genetic mutations leading to atypical morphogenesis of the vascular tissue. VMs are composed of dysplastic vessels lined by an endothelial layer. Although present at birth, they may not be detected until later in life. Malformations never regress and grow proportionally with the individual. Abnormal tissue may expand due to trauma, hormonal changes or infection [1,2,3].

Pathological angiogenesis may also lead to vascular tumours such as benign haemangiomas and malignant angiosarcomas. Haemangiomas are mostly seen in infants. Endothelial cells are indistinguishable from normal human endothelium, but a local imbalance between angiogenic stimulators and inhibitors in the development may lead to higher morbidity and mortality through haemorrhage. Angiosarcomas’ clinical presentation can often imitate benign lesions; however, they have a high tendency to metastasize. Those tumours are poorly responsive to chemotherapy and radiation; therefore, early diagnosis is crucial to a good prognosis [4].

VMs mainly occur in the upper part of the human body. Over 40% of VMs localise in the head and neck. The most common areas are the eyelid, lip, chin, buccal space, parapharyngeal space, submandibular space and neck. Clinical findings contain well-defined, soft and compressible masses without pulsation (if venous malformations). They may also involve each body part and enlarge during the Valsalva manoeuvre [5].

Venous malformations, similar to other VMs, are present at birth. They are the most common type of VMs (approximately two-thirds of all), affecting 1–4% of individuals [2,6]. Lymphatic malformations (LMs) affect 1.2–2.8% of the population and, in most cases, occur in the head and neck region as lymphatic venous malformations (LVMs) [7]. Capillary malformations can be found in only 0.5% of the population [2].

In the general ear, nose and throat (ENT) population, the incidence of venous thromboembolism ranges between 0.1% and 1.6%, which is still an uncommon site of this condition [8]. It is extremely rare in this body region, but thrombosis may also occur in these abnormal vessels, imitating a tumour [9].

We discuss a series of cases of thrombosis in the parotid region, initially mimicking parotid gland tumours, to enable others to conduct an accurate diagnostic process and help to choose the right treatment.

## 2. Case Presentations

The presented patients were diagnosed at the University Clinical Center in Gdansk, Poland, for the presence of a nodule in the parotid region. Because of the non-specific nature of symptoms and relatively short clinical history, prompt diagnostics were ordered due to the risk of neoplastic disease.

### 2.1. Case 1

A 57-year-old female was admitted to the ENT outpatient clinic because of a painless nodule of the left parotid region that was observed for two months. In physical examination, enlargement and increased firmness of the left parotid gland were noted. The patient was scheduled for an ultrasound (US) examination. The first US revealed a suspicious polycyclic hypoechoic focal lesion (38 mm × 14 mm) in the superficial lobe of the parotid gland (Figure 1a). A fine-needle aspiration (FNA) was performed, and the cytology examination revealed blood cells only. The cytology report suggested a highly vascularised tumour.

Magnetic resonance imaging (MRI) was ordered for further differentiation of the lesion. MRI revealed a solitary lesion measuring 18 mm × 9 mm × 35 mm, with a bright hyperintense centre and a dark hypointense rim in T1-weighted images (Figure 1b), a mixed, streaky structure in T2-weighted images (Figure 1c) and with free diffusion restriction (ADC = 0.77 × 10^−3^ mm^2^/s) in diffusion-weighted imaging (DWI) (Figure 1e,f).

After reassessment of the MRI examination, a diagnosis of thrombosis was suggested. Additionally, a hyperintense, ovoid lesion in T2-weighted images was spotted in the proximity of the left temporomandibular joint (Figure 1d). Follow-up ultrasound showed the lesion in the superficial lobe of the left parotid gland, adjacent to the retromandibular vein, with slow flow around the lesion in colour Doppler examination. The widened vessel lumen was hypoechoic and heterogeneous. Considering all the workup, the most likely diagnosis was thrombosis in a LVM.

The patient was placed under ultrasound surveillance. In the 6-month follow-up, there were no signs of the previously observed thrombus in the widened vein. The vein was compressible, with slower blood flow in the lateral portion of the LVM (Figure 1g,h).

### 2.2. Case 2

A 62-year-old female with a known history of LVMs of the submandibular space and floor of the oral cavity came to the clinic due to a newly spotted, painless nodule in the right parotid region. About forty years before, in the surrounding of the temporomandibular joint, an LVM was surgically partly removed.

The patient underwent an US examination (Figure 2a). An ovoid lesion of 14 mm in the largest diameter was found in the anterior process of the right parotid gland. The lesion had mixed echogenicity (mainly hypoechoic) with marginal irregularities. Due to an atypical presentation, FNA was ordered. In the cytology examination, only blood cells were found. Subsequently, a contrast-enhanced MRI was performed. Scans showed a mixed intensity lesion in T1-weighted images (Figure 2b), hyperintense in T2-weighted images (Figure 2c) and free diffusion restriction (ADC = 0.88 × 10^−3^ mm^2^/s) in DWI (Figure 2d,e). After contrast agent administration, a central enhancement in the lesion was noted. Moreover, an extensive LVM in the right sublingual, submandibular, masseteric and parotid space was noted. Finally, the lesion was interpreted as a thrombus in LVM.

A year after diagnosis of thrombosis, the patient was qualified for percutaneous sclerotherapy of the malformation in the anterior part of the superficial lobe of the right parotid gland. In the 1-year follow-up MRI, the lesion’s size had decreased, and there were no signs of previously observed thrombus and diffusion restriction. In the gradient echo sequence (GRE), blood products were spotted.

### 2.3. Case 3

A 43-year-old male with a history of headaches and paraesthesia of the right auricular region was admitted to the clinic due to a focal lesion in the right parotid gland. In the US examination, a round, heterogenous and hypoechoic lesion was visible (Figure 3a). Later, in FNA, only blood cells were found. Due to an inconclusive cytology result, an MRI was ordered. The examination revealed a solitary, well-circumscribed, lobulated lesion in the upper pole of the parotid gland, between the superficial and deep lobe. The central part of the lesion was hypointense in T1-weighted images (Figure 3b), hyperintense in T2-weighted images (Figure 3c), and the lesion’s rim included hyperintense small foci in T1-weighted images. Dynamic contrast-enhanced MRI showed the curve type A enhancement of the central part of the lesion and a lack of contrast enhancement in the rim. Diffusion restriction in the lesion was noted (ADC = 0.87 × 10^−3^ mm^2^/s) (Figure 3d,e). In a retrospective assessment of the previous MRI scan acquired two years before current imaging, a much smaller lesion was noticeable in the exact location (Figure 3f). Due to size progression, diffusion restriction in MRI and non-diagnostic first cytology, another FNA was ordered. The second FNA material from the needle was additionally secured for a cell block. Cytology again revealed only blood cells, which was interpreted as a non-diagnostic result. Due to clinical symptoms and ambiguous imaging, a complete left parotidectomy was performed. After removing an unaffected superficial lobe and repositioning the branch of the facial nerve, a surgeon removed a dark coloured, encapsulated tumour (2 × 1 cm) from the deep lobe of the salivary gland. The appearance was typical for vascular tumours. A histopathological exam revealed vascular malformation with glomeruloid vessels, accompanied by extensive thrombosis.

### 2.4. Case 4

A 41-year-old female nurse was admitted to the clinic due to a painful nodule in the left parotid region, near the angle of the mandible. US examination revealed a lesion localised on the left masseter. The lesion appeared three months earlier and presented as a subcutaneous firm nodule measuring about 1 cm. The lesion was oval, non-compressive and hypoechogenic (8 mm × 3 mm), with no flow in Doppler examination (Figure 4a). After four days of empiric antibiotic therapy, the size and pain of the lesion decreased. A 2-week follow-up showed a decrease in the lesion’s size and a change of echogenicity to hyperechogenic. The lesion was still non-compressible, with weak flow in the Doppler examination around the structure (Figure 4b). FNA revealed blood cells only. The ordered MRI showed an oval nodule on the left masseter, hypointense in T1-weighted images (Figure 4c), and hyperintense in T2-weighted images (Figure 4d), measuring up to 5 mm × 8 mm × 10 mm. The central part of the lesion was hyperintense in T1 and hypointense in T2-weighted images with free diffusion restriction in DWI (Figure 4d,e). Contrast-enhanced MRI sequences revealed marginal enhancement (Figure 4f).

A diagnosis of thrombosis in the masseter’s vein was made based on a performed examination. In a follow-up US examination 3 months later, the lesion had a stable picture.

## 3. Discussion

The parotid region is not a typical location for VMs, with about 50 cases reported [10]. Achache et al. demonstrated that, among 614 parotidectomy cases identified, there were only 10 cases of parotid VMs (1.6%). Moreover, 90% of the VMs were found in females, all of them were unilateral, and 60% were situated in the superficial lobe of the parotid gland. Even if clinical presentation was typical for a benign lesion, none of the imaging techniques—including US, CT or MRI—allowed the proper diagnosis of a VM [11].

Due to the abnormal morphology of VMs, there is pathological turbulent blood or lymph flow. Localised intravascular coagulopathy is the primary characteristic of low-flow venous malformations. The risk group mainly includes people with extensive and more profound VMs [12]. Phleboliths are common in VMs, but the real diagnostic challenge is identifying VMs forming a thrombus without characteristic features. We searched the PubMed and Google Scholar databases and found less than 20 cases of thrombosis in VMs of the parotid region. Less than half of them had no phleboliths present.The differential diagnosis of deep lesions includes lymphadenopathy, benign or malignant tumours, and cystic lesions [13]. Exclusion of malignancy in the parotid region becomes a priority for the clinician. The extremely rare incidence of thrombosis in this area, and the possible unclear clinical picture of this pathology, presents a diagnostic challenge. With the growing capabilities of radiology, the accuracy of non-invasive diagnostic methods of the lesion in the parotid rises. The proper diagnosis is essential as it determines the treatment that the Laryngologist will proceed with.

We presented a wide variety of different lesions with thrombosis in the parotid region. Clinical and radiological symptoms were ambiguous. In three of the four cases, there was no biased medical history. Only one patient had a history of LVM in the head and neck area, which did not affect the diagnostics. The remaining patients had a short medical history of symptoms.

Based on the findings in US examinations, FNA was ordered. In all of the cases, cytology revealed blood cells only. The cytology results were interpreted as non-diagnostic—category I according to the Milan System for Reporting Salivary Gland Cytopathology—with a potential risk of malignancy of 25% [14]. In our experience, blood cells in cytology are often observed in Warthin tumour biopsies.

In the MRI, three of the four lesions presented with hyperintense foci in T1-weighted images, which might suggest blood products. However, this sign is non-specific and may represent lipids in Warthin tumours [15], high protein content, haemorrhagic changes in some malignancies [16] or haemorrhagic foci after FNA. Moreover, a thrombus may change signal characteristics in T1- and T2-weighted images with time, due to haemoglobin conversion [17]. In all the lesions, there was free diffusion restriction in DWI related to the presence of the thrombus. Diffusion restriction can be observed within pus collection, in the thrombus or in malignancies [18,19]. Thrombus and highly cellular malignancies can restrict diffusion significantly. Unless cystic, malignancies appear as areas of fairly homogenous diffusion restriction. In cystic areas of malignancy, no restriction will be noted. In the presented cases, the hyperintense foci in T1-weighted images did not exactly match the areas of diffusion restriction, which contributed to the ambiguous presentation.

In one case, the change of echogenicity of the lesion in the US was noted—a typical sign of the evolution of thrombus.

To properly diagnose pathology in the parotid region, the most accessible US with colour Doppler can be applied. VMs have a typical image of compressible lesions—mostly anechoic, with possible septa, with fast, low, or no-flow, depending on the type of lesion. Contrast-enhanced CT has limited effectiveness in the characterisation of parotid lesions and VMs. MRI ensures high soft-tissue resolution and allows the assessment of diffusion restriction, dynamic contrast-enhanced examination, and detection of blood degradation products in gradient echo sequences and DWI. VMs appear as hypointense on T1-weighted images, and hyperintense on T2-weighted images, and as lesions with lobulated margins, without diffusion restriction, often transspacial or multispatial. LMs and LVMs can have fluid–fluid levels in T2-weighted images. In the event of thrombosis, a high signal in T1-weighted images, diffusion restriction and a blooming artefact in the gradient echo sequence may appear within VM. Limitations of the gradient echo sequence include possible intratumoral haemorrhage, the presence of calcifications, and also vascular phleboliths forming from an old thrombus [20].

Due to the lack of standards for treatment of thrombosis in VMs on the head and neck, choosing the appropriate strategy was challenging. The selected management in the presented cases was varied—one lesion was surgically removed because of clinical symptoms and size progression, and one patient was put under US surveillance due to stable clinical and radiological picture; in the last two cases, spontaneous regression of thrombosis was observed.

The treatment for VMs thrombosis on the head and neck should be individualised, and an important factor affecting the decision is the existance of thrombosis elsewhere.

Dompmartin et al. reported patients with localised intravascular coagulopathy, causing pain and thrombosis. The authors stated that elevated D-dimer levels were statistically insignificant to localised intravascular coagulopathy in VMs on the head and neck (31% with increased D-dimer level) [12]. They point out that the small size and primarily superficial location of VMs on the head and neck corresponds with a more likely negative D-dimer laboratory result. Dompmartin et al. explained that antiplatelet drugs such as aspirin have low efficacy because platelets are not involved in localised intravascular coagulopathy. Moreover, this drug raises the risk of complications, such as bleeding. The efficient treatment is low-molecular-weight heparin (LMWH), which can stop the pain caused by coagulopathy, as observed in all of his patients.

On the other hand, Soudet et al. reported using antiplatelet and antithrombotic drugs in patients with venous thrombosis in facial and non-facial VMs. All of the patients noticed a reduction in symptoms and complete thrombosis resolution. However, the authors point out that this treatment was started despite the lack of recommendations for this type of thrombosis [9]. Moreover, Richter and Braswell demonstrated that many patients respond to low-dose aspirin (81 mg), which may be a suitable alternative to low-molecular-weight heparin [21].

Other authors suggest that the use of dabigatran (compared with low-molecular-weight heparin) [22] or rivaroxaban alone [23], can be successful in controlling signs of coagulopathy and avoiding complications in VMs of the head and neck.

Nakano and Zeinati point out the need to create anticoagulation therapy initiation criteria. It can be based on commonly evaluated biochemical markers—such as D-dimer and fibrinogen levels—taking into account types of vessel ectasia and the clinical condition of the patient [24]. According to the review published by Cramer et al., clinicians should use the Caprini system, which allows for the stratification of the risk of thromboembolic events in ENT patients [25].

Furthermore, patients with head and neck VMs may take precautions by keeping their heads slightly elevated during sleep.

## 4. Conclusions

The most common localization of VMs is the head and neck region. Abnormal morphology of vessels increases the risk of forming a thrombus. Due to the often atypical clinical and radiological presentation of the lesions—mimicking both benign and malignant tumours—wide differential diagnosis is necessary. Ambiguous imaging in conjunction with purely blood cells in cytology should result in including thrombosis in VMs in the differential diagnosis. Treatment of thrombosis in VMs differs from management of malignant tumours. Therefore, including VMs’ thrombosis in differential diagnosis in the right moment may spare costly diagnostics, unnecessary operations and stress for the patient.

## Figures and Tables

**Figure 1 diagnostics-12-02236-f001:**
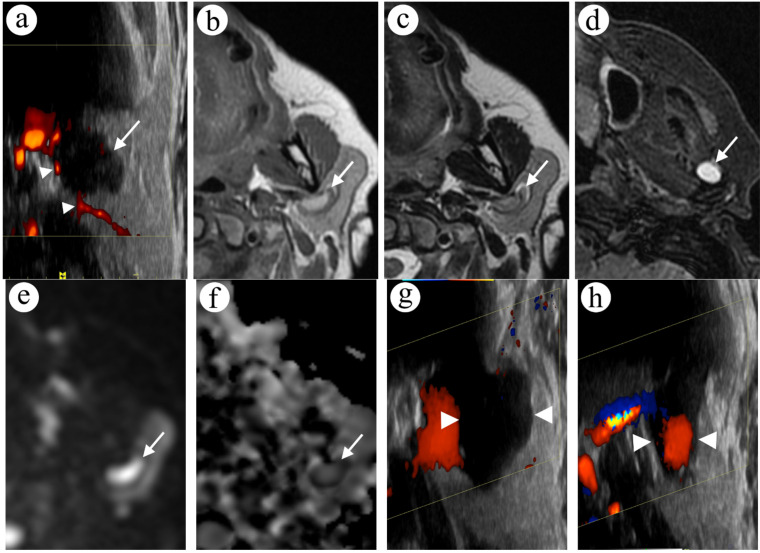
A focal lesion in the superficial lobe of the left parotid gland in US and MRI; all US images are rotated 90 degrees to match the anatomy in MR images. A polycyclic, hypoechoic and heterogenous lesion (arrow) is visible in US examination (**a**) with some vascular flow in the periphery of the lesion (arrowheads). In the turbo spin echo sequence in axial T1-weighted (**b**) and T2-weighted images (**c**), an ovoid lesion with a rim can be spotted (arrow). Additionally, in the fat-saturated short TI inversion recovery sequence in the axial T2-weighted image (**d**), a hyperintense lesion is adjacent to the left temporomandibular joint (arrow). In DWI (**e**) and the ADC map (**f**), the lesion shows diffusion restriction in the rim (arrow). In the follow-up US examination without (**g**) and with compression (**h**), the lesion is deformable (arrowheads).

**Figure 2 diagnostics-12-02236-f002:**
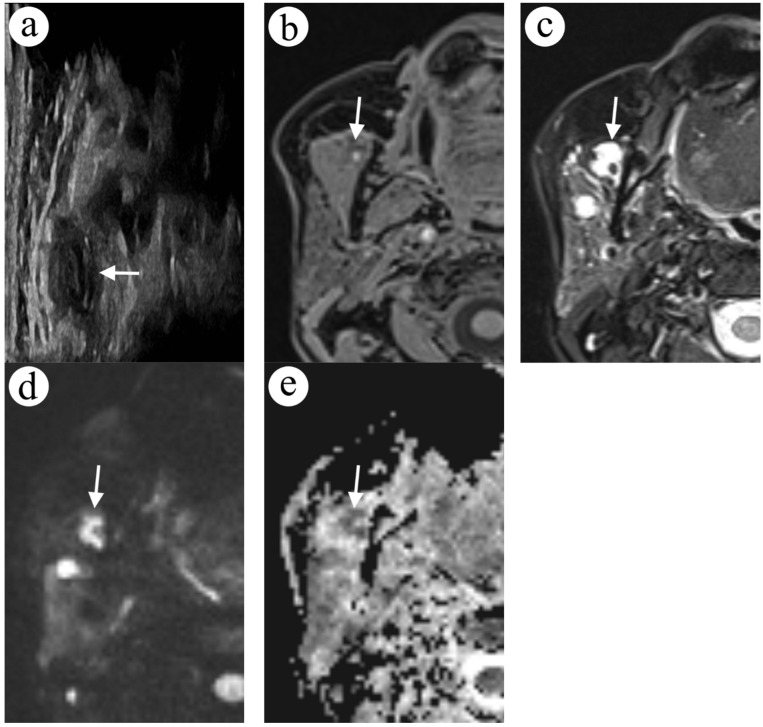
A focal lesion in the anterior process of the right parotid gland in US and MRI; US image is rotated 90 degrees to match the anatomy in MR images. The lesion (arrow) has mixed echogenicity in the US examination (**a**). In the fat-saturated gradient echo sequence (VIBE Dixon) in the axial T1-weighted image, the lesion has mixed intensity with hyperintense foci (**b**). In the fat-saturated turbo spin echo sequence in the axial T2-weighted image, the lesion (arrow) presents as hyperintense (**c**) and shows a rim of diffusion restriction (arrow) in readout-segmented DWI (**d**) and the ADC map (**e**).

**Figure 3 diagnostics-12-02236-f003:**
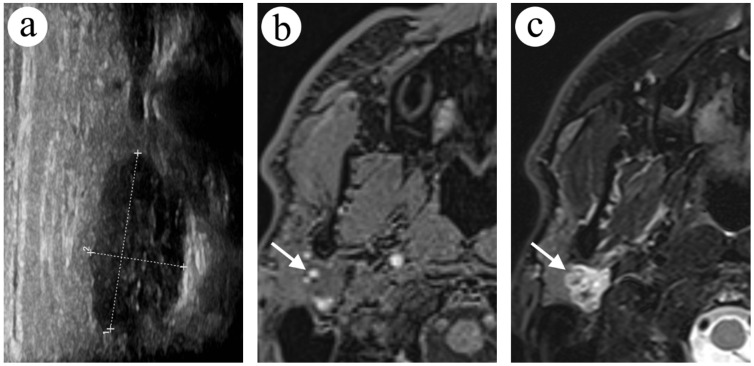
A focal lesion between the right parotid gland’s superficial and deep lobe in US and MRI. In the coronal plane in the US examination, the lesion was heterogenous and hypoechoic (**a**). In the fat-saturated gradient echo sequence (VIBE Dixon) in the axial T1-weighted image (**b**), a lobulated lesion with hyperintense foci in the periphery is visible (arrow). In the fat-saturated turbo spin echo sequence in the axial T2-weighted image, the lesion (arrow) was strongly hyperintense (**c**). The lesion presented diffusion restriction in the rim in readout-segmented DWI (**d**) and ADC maps (**e**). In the fat-saturated turbo spin echo sequence in the axial T2-weighted image in the previous examination (**f**), the lesion (arrow) was much smaller.

**Figure 4 diagnostics-12-02236-f004:**
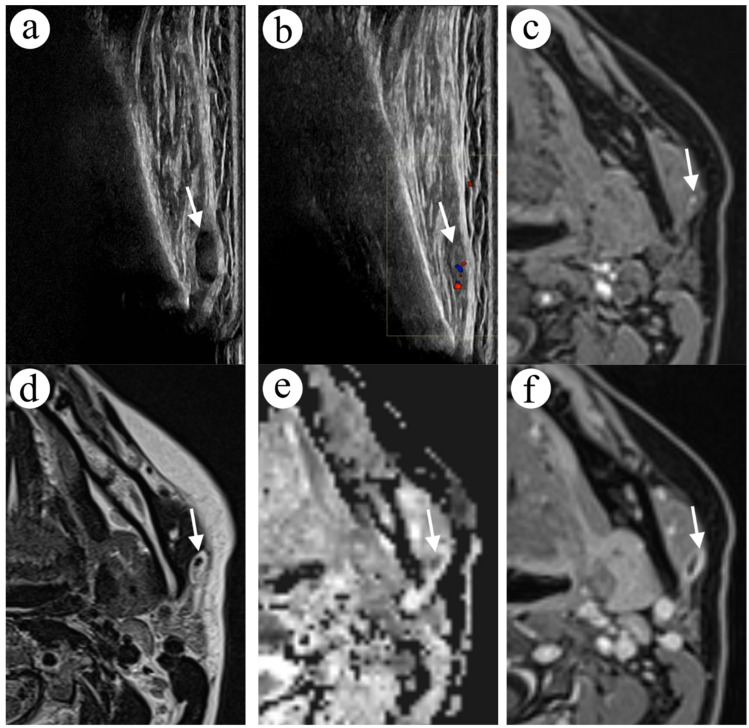
A focal lesion in the left parotid region, adjacent to the posterior border of the masseter muscle US image, rotated 90 degrees to match the anatomy in MR images. In the first US examination, the lesion presented as hypoechoic (arrow, **a**) and changed to hyperechoic in the second US examination (arrow, **b**). In the fat-saturated gradient echo sequence (VIBE Dixon) in the axial T1-weighted image (**c**), the lesion was hypointense with a hyperintense focus in the centre (arrow). In the turbo spin echo sequence in the axial T2-weighted image (**d**), the lesion had a hypointense focus in the centre (arrow). The lesion restricted diffusion in DWI (**e**) and showed peripheral enhancement (arrow) in the fat-saturated gradient echo sequence (VIBE Dixon) in the axial T1-weighted image (**f**).

## Data Availability

Not applicable.

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
