# Peer review of "Vascular Abnormalities of the Parotid Region: An Uncommon Presentation of a Common Condition—A Case Series"

_diagnostics, 2022, doi:10.3390/diagnostics12092236_

Round 1

Reviewer 1 Report

1) Abstract. L11-16. We present a case series of 4 patients reported to the outpatient clinic with a unilateral  nodule in the parotid region. Ambiguous initial diagnostics related to the unclear anamnesis and short medical history failed to raise the correct diagnosis. The suspicion of a neoplastic disease resulted in profound diagnostics, including repeated cytology, ultrasound and magnetic resonance  examination. Combining all acquired information and follow-up or a histopathologic examination  facilitated the final diagnosis. In all cases a thrombosis was diagnosed. Underneath, we analyse the  diagnostic process and the associated difficulties. Could you please add a background?

2) Abstract. L11-16. We present a case series of 4 patients reported to the outpatient clinic with a unilateral  nodule in the parotid region. Ambiguous initial diagnostics related to the unclear anamnesis and short medical history failed to raise the correct diagnosis. The suspicion of a neoplastic disease resulted in profound diagnostics, including repeated cytology, ultrasound and magnetic resonance  examination. Combining all acquired information and follow-up or a histopathologic examination  facilitated the final diagnosis. In all cases a thrombosis was diagnosed. Underneath, we analyse the  diagnostic process and the associated difficulties.  Could you please divide the abstract in different section for example background, case series, conclusion.

3) 1. Introduction L20-25. Vascular malformations (VMs) belong to vascular abnormalities resulting from  genetic mutations leading to atypical morphogenesis of the vascular tissue. VMs are  composed of dysplastic vessels lined by an endothelial layer. Although present from birth, they may not be detected until later in life. Malformations never regress and grow  proportionally with the individual. Abnormal tissue may expand due to trauma, hormonal changes and infection [1–3]. Please add a brief paragraph on different on vessels and endothelial layer pathologies:

A)  Hemangiomas, angiosarcomas, and vascular malformations represent the signaling abnormalities of pathogenic angiogenesis. Curr Mol Med. 2009 Nov;9(8):929-34. doi: 10.2174/156652409789712828. 

B) Correlations between blood perfusion and dermal thickness in different skin areas of systemic sclerosis patients. Microvasc Res. 2018 Jan;115:28-33. doi: 10.1016/j.mvr.2017.08.004. 

4) Introduction. We discuss a series of cases focused on diagnosing and treating thrombosis in the  parotid region. Could you please improve the description of study aim?

5) 3. Discussion  L173-175.  The parotid region is not a typical location for VMs. There have been reported about 50 cases of VMs in the parotid region [9]. Achache et al., in their 10-year study, report that  parotid VMs constitute about 1.6% of all parotid tumour cases [10]. Could you please improve this sentence and add some information on this interesting study.

6) 4. Conclusions L265-272. VMs are relatively frequent, especially in the head and neck area. Their abnormal morphology increases the risk of forming a thrombus. When thrombosis in VMs occurs in  the parotid region, it may have an unclear clinical and radiological presentation. Such an  image can imitate both benign and malignant tumours. Ambiguous imaging in conjunction with blood cells in cytology should result in including thrombosis in VMs in the differential diagnosis. Due to the rarity of this condition, treatment is individualised, usually symptomatic. In selected cases, diagnostics for other coagulation risk factors may be considered. Please improve this paragraph and underline the clinical implication of these case series.

Author Response

Dear Reviewer,

Thank you for your time and valuable notices. Below you will find our answers.

1) Abstract. L11-16. We present a case series of 4 patients reported to the outpatient clinic with a unilateral  nodule in the parotid region. Ambiguous initial diagnostics related to the unclear anamnesis and short medical history failed to raise the correct diagnosis. The suspicion of a neoplastic disease resulted in profound diagnostics, including repeated cytology, ultrasound and magnetic resonance  examination. Combining all acquired information and follow-up or a histopathologic examination  facilitated the final diagnosis. In all cases a thrombosis was diagnosed. Underneath, we analyse the  diagnostic process and the associated difficulties. Could you please add a background?

A clinical background was added.

2) Abstract. L11-16. We present a case series of 4 patients reported to the outpatient clinic with a unilateral  nodule in the parotid region. Ambiguous initial diagnostics related to the unclear anamnesis and short medical history failed to raise the correct diagnosis. The suspicion of a neoplastic disease resulted in profound diagnostics, including repeated cytology, ultrasound and magnetic resonance  examination. Combining all acquired information and follow-up or a histopathologic examination  facilitated the final diagnosis. In all cases a thrombosis was diagnosed. Underneath, we analyse the  diagnostic process and the associated difficulties.  Could you please divide the abstract in different section for example background, case series, conclusion.

The abstract was divided into background, case reports, results and conclusions as suggested.

3) 1. Introduction L20-25. Vascular malformations (VMs) belong to vascular abnormalities resulting from  genetic mutations leading to atypical morphogenesis of the vascular tissue. VMs are  composed of dysplastic vessels lined by an endothelial layer. Although present from birth, they may not be detected until later in life. Malformations never regress and grow  proportionally with the individual. Abnormal tissue may expand due to trauma, hormonal changes and infection [1–3]. Please add a brief paragraph on different on vessels and endothelial layer pathologies.

  1. A) Hemangiomas, angiosarcomas, and vascular malformations represent the signaling abnormalities of pathogenic angiogenesis. Curr Mol Med. 2009 Nov;9(8):929-34. doi: 10.2174/156652409789712828. 
  2. B) Correlations between blood perfusion and dermal thickness in different skin areas of systemic sclerosis patients. Microvasc Res. 2018 Jan;115:28-33. doi: 10.1016/j.mvr.2017.08.004. 

Description of other vascular lesions was added according to article (A).

4) Introduction. We discuss a series of cases focused on diagnosing and treating thrombosis in the  parotid region. Could you please improve the description of study aim?

We added the aim of the study in the last paragraph of the Introduction.

5) 3. Discussion  L173-175.  The parotid region is not a typical location for VMs. There have been reported about 50 cases of VMs in the parotid region [9]. Achache et al., in their 10-year study, report that  parotid VMs constitute about 1.6% of all parotid tumour cases [10]. Could you please improve this sentence and add some information on this interesting study.

The sentence was rephrased and additional information form the study was introduced in the paragraph.

6) 4. Conclusions L265-272. VMs are relatively frequent, especially in the head and neck area. Their abnormal morphology increases the risk of forming a thrombus. When thrombosis in VMs occurs in  the parotid region, it may have an unclear clinical and radiological presentation. Such an  image can imitate both benign and malignant tumours. Ambiguous imaging in conjunction with blood cells in cytology should result in including thrombosis in VMs in the differential diagnosis. Due to the rarity of this condition, treatment is individualised, usually symptomatic. In selected cases, diagnostics for other coagulation risk factors may be considered. Please improve this paragraph and underline the clinical implication of these case series.

The paragraph was rephrased and clinical implication was emphasized.

Reviewer 2 Report

Subject of the presented paper is a case series of vascular abnormalities mimicking a parotid tumour. For the ENT specialist (my profession!), the case reports are further examples of the wide variety of parotid neoplasms. The most significant value concerns differential diagnosis.

Nevertheless, the manuscript should be reworked carefully before being published.

Formal comments:

Language must be improved be a native speaker.

The authors introduce a high number of abbreviations. This makes it difficult to read the text. Abbreviations should be limited or a list of abbreviations is needed.

Concerning the structure of the presented text:

·         The case reports presented by the authors are interesting, the figures are of good quality underlining the written text. However, the authors present their cases for three times (case report, footer of the figure and within table 1). Within case presentations repetitions should be carefully avoided. 

·         Length of the text is critical aspect of the “discussion” as well (>2 pages totally!). 

·         Discussion of diagnostic methods (US/MRI; ll196-231) should be summerized within one chapter.

·         The value of chapter 3.2 is not comprehensible to me: there is no reference to (differential-) diagnostic. Moreover, the general aspects presented concerning thrombosis would not be applied in this location.

Content (Discussion):

It should be pointed out that the usual and concluding diagnostic of a parotid lump comprises clinical examination and sonography, followed by surgery.

FNA cytology may be a standard in some institutions. On the background of their results the authors should discuss the value and the limitations of the method.

MR imaging (CT) is applied in care cases, predominantly if the tumour is located in the deep part of the gland.

However, presented MRI results may be of special interest for the radiologist. As I´m ENT surgeon I cannot assess this.

Author Response

Dear Reviewer,

Thank you for your time and valuable notices. Below you will find our answers.

  1. Language must be improved by a native speaker.

A native speaker revised the manuscript (Daniel Holford, [email protected]).

  1. The authors introduce a high number of abbreviations. This makes it difficult to read the text. Abbreviations should be limited or a list of abbreviations is needed.

Some abbreviations have been removed (DD, GRE, LIC, WI ) and a list of abbreviations has been added.

  1. The case reports presented by the authors are interesting, the figures are of good quality underlining the written text. However, the authors present their cases for three times (case report, footer of the figure and within table 1). Within case presentations repetitions should be carefully avoided.

The table has been removed.

  1. Length of the text is critical aspect of the “discussion” as well (>2 pages totally!).

The Discussion section has been altered and shortened.

  1. Discussion of diagnostic methods (US/MRI; ll196-231) should be summarized within one chapter.

The discussion of radiological methods has been moved to the chapter concerning imaging of the patients.

  1. The value of chapter 3.2 is not comprehensible to me: there is no reference to (differential-) diagnostic. Moreover, the general aspects presented concerning thrombosis would not be applied in this location.

This chapter has been shortened and merged with paragraph concerning treatment of presented patients. Additionally, an explanation for the discussion was added in the beginning of the paragraph.

  1. It should be pointed out that the usual and concluding diagnostic of a parotid lump comprises clinical examination and sonography, followed by surgery.

We have to agree that diagnostics of parotid tumours can be based solely on clinical history and examination, and ultrasonography. However, in our institution after clinical examination usually an ultrasonography is performed. If ultrasonography confirms focal lesion a FNA is ordered. If cytology is ambiguous or carcinoma is suspected, MRI usually follows.

It is worth mentioning that ‘when FNA cytology is used to subclassify the neoplasm, the accuracy ranges widely from 48% to 94%’ [Kala C, Kala S, Khan L. Milan System for Reporting Salivary Gland Cytopathology: An Experience with the Implication for Risk of Malignancy. J Cytol. 2019 Jul-Sep;36(3):160-164. doi: 10.4103/JOC.JOC_165_18].

  1. FNA cytology may be a standard in some institutions. On the background of their results the authors should discuss the value and the limitations of the method.

Additional information on FNA was added.

  1. MR imaging (CT) is applied in rare cases, predominantly if the tumour is located in the deep part of the gland. However, presented MRI results may be of special interest for the radiologist.

In all discussed cases the clinical picture was unclear and FNA ambiguous. Therefore, further diagnostics was performed, including MR and follow-up ultrasound.